

# Mulch and groundcover effects on soil temperature and moisture, surface reflectance, grapevine water potential, and vineyard weed management

Christina M. Bavougian and Paul E. Read

Department of Agronomy & Horticulture, University of Nebraska—Lincoln, Lincoln, NE, United States of America

## ABSTRACT

The objectives of this research were to identify alternatives to glyphosate for intra-row (under-trellis) vineyard floor management and to evaluate the potential for intra-row and inter-row (alleyway) groundcovers to reduce vegetative vigor of 'Marquette' grapevines (*Vitis* spp.) in a southeast Nebraska vineyard. The experiment was a randomized factorial design with five intra-row treatments (crushed glass mulch [CG], distillers' grain mulch [DG], creeping red fescue [CRF], non-sprayed control [NSC], and glyphosate [GLY]) and three inter-row treatments (creeping red fescue [CRF], Kentucky bluegrass [KB], and resident vegetation [RV]). Treatments were established in 2010–2011 and measurements were conducted during 2012 and 2013 on 5- and 6-year-old vines. Soil temperatures were mostly higher under mulches and lower under intra-row groundcovers, compared to GLY. Weed cover in CG, DG, and CRF treatments was the same or less than GLY. At most sampling dates, inter-row soil moisture was lowest under KB. Intra-row soil moisture was highest under DG mulch and lowest under CRF and NSC; CG had the same or lower soil moisture than GLY. Surprisingly, we did not detect differences in mid-day photosynthetically active radiation (PAR) reflectance, despite visual differences among the intra-row treatments. Mid-day vine water potential did not differ among treatments. We concluded it is not necessary to maintain a bare soil strip under established vines in this region, where soil fertility and moisture are non-limiting.

# INTRODUCTION

Midwestern vineyards are often situated on land that has been converted from row-crop agriculture. Due to grapevines' (*Vitis* spp.) perennial nature, vineyards require much less soil disturbance through cultivation than conventionally grown corn (*Zea mays* L.), soybean [*Glycine max* (L.) Merr.] and wheat (*Triticum aestivum* L.). The soil-conserving practice of maintaining permanent sod cover between vineyard rows is popular with grape growers in the upper Midwest and contributes to the sustainability of vineyard agroecosystems.

Corresponding author
Christina M. Bavougian,
cbavougian2@unl.edu

However, the "grower standard" of spraying glyphosate or other herbicides under the vine rows to keep a weed-free bare strip under the trellis is problematic.

Overuse of glyphosate in agriculture contributes to environmental problems such as evolution of resistant weed populations and economic problems including vine die-back due to trunk damage from overspray. Glyphosate treatments decreased cold hardiness and increased winter trunk cracking in several woody species (*Daniel, Mathers & Case, 2009*; *Rosenberger et al., 2010*); we propose that trunk splitting and vine dieback in vineyards previously assumed to be frost/freeze damage could be caused by glyphosate overspray. Mulches and groundcovers have been identified as potential alternatives to herbicides in vineyards. Their benefits include improved soil structure, reduced soil erosion and increased water infiltration (*DeVetter, Dilley & Nonnecke, 2015*).

In areas with ample rainfall, soils high in fertility and organic matter may produce an overabundance of grapevine vegetative growth. High vigor, or capacity for growth, is demonstrated in vines whose large, dense canopies result in low light penetration (*Byrne & Howell, 1978*). The condition of excessive vigor causes many problems. Self-shading is the most dire consequence; overly vigorous vines may lead to reductions in fruit yield and quality (*Shaulis, Amberg & Crowe, 1966*), cold tolerance (*Byrne & Howell, 1978*), and the next growing season's fruitfulness (*Shaulis, Amberg & Crowe, 1966*). Decreased fruitfulness causes subsequent yield reduction, which increases vegetative vigor–contributing to the cyclic nature of this problem (*Byrne & Howell, 1978*). The canopy density of high vigor vines creates unfavorable canopy microclimate conditions; reduced air movement and high humidity favor disease incidence (*Smart & Robinson, 1991*). Hedging, summer pruning, and leaf removal become necessary under conditions of excessive vine vigor, while normal activities such as dormant pruning and shoot positioning take much longer on overly vigorous vines (*Smart & Robinson, 1991*). Many reports show the potential for groundcovers to reduce excessive vine vigor (*Giese et al., 2014*; *Giese et al., 2015*; *Hatch, Hickey & Wolf, 2011*; *Tesic, Keller & Hutton, 2007*) which can reduce labor requirements and improve fruit yield and quality. Some mulches increase the amount of solar radiation that is reflected, which may improve the light environment within the canopy (*Smart, Smith & Winchester, 1988*).

This project compared five intra-row (under-trellis) management strategies: crushed glass mulch, dried distillers' grain mulch, creeping red fescue (*Festuca rubra* L.) groundcover, glyphosate (bare soil) and non-sprayed control. Due to prohibitively high transportation costs associated with shipping glass to recycling facilities, most municipal waste glass is sent to local landfills instead (*Niesse, 2015*; *Hill, 2016*). Horticultural applications such as vineyard mulch can remove glass from the waste stream. Dried distillers' grains, a co-product of the ethanol production process, is often utilized in conventional livestock feed programs. A surplus of distillers' grains may become available as a result of projected increases in ethanol production. Researchers have recently discovered this material's potential to reduce weed germination and emergence (similar to pre-emergence herbicides) when applied to the soil surface (*Boydston, Collins & Vaughn, 2008*). Our research evaluated the previously-mentioned intra-row groundcovers, as well as inter-row groundcover treatments: creeping red fescue, Kentucky bluegrass (*Poa pratensis* L.) and

resident vegetation which was dominated by orchard grass (*Dactylis glomerata* L.). Grass species differ in root structure and water use efficiency (*Frandsen, De Kroon & Berendse, 1998*); therefore, they could affect grapevine growth and water status differently.

The objectives of this research were to identify alternatives to synthetic chemicals used in vineyard floor management, and to reduce grapevine vegetative vigor using intra-row groundcovers and different inter-row groundcover species. In this paper, we compare the inter-row and intra-row vineyard floor management treatment effects on soil temperature, surface reflectance, soil moisture, vine water potential and weed control. Grower recommendations and practical interpretation of the data are included.

## MATERIALS & METHODS

### Site description

Research for this study was conducted at Fox Run Farms, a diversified family farm in Brainard, NE (latitude 41.18N; longitude −97.00W) from August 2010 to November 2013. The soil is a Hastings silty clay loam with 4% organic matter. The two-acre vineyard was planted in spring 2007 with own-rooted 'Marquette' vines, a cold-hardy and disease-resistant red wine grape hybrid from the University of Minnesota's breeding program released in 2006 (*Minnesota, 2017*). Rows were oriented north-south with vines 2.4 m apart in rows 3 m apart. The vineyard was situated on a south-facing slope of approximately 3 percent. Vines were trained to cordons 0.9 m high with vertical shoot positioning.

### Treatments and experimental design

The experimental layout was a completely randomized factorial design with four replications. Five intra-row treatments (recycled crushed glass mulch (CG), dried distillers' grain mulch (DG), 'Boreal' creeping red fescue (CRF), non-sprayed control (NSC) and glyphosate (GLY)) and three inter-row (alleyway) treatments ('Boreal' creeping red fescue (CRF), 'Park' Kentucky bluegrass (KB), and resident vegetation (RV) which was dominated by orchard grass) were established in the research plots. The NSC plots were initially seeded with 'Dalkeith' subterranean clover, which did not survive well over winter and were thereafter maintained as non-sprayed control.

The treatment combinations were applied to three-vine plots, which included the intra-row strip and inter-rows on both sides of the vines, with a buffer row between adjacent plots. Intra-row treatments were applied to a 0.9-m-wide area under the vines, while 2.1-m-wide strips of CRF, KB, and RV were maintained in the inter-rows. Measurements were taken on and around the middle vine in each plot, to eliminate edge effects. Inter-row groundcover plots were prepared by treating the resident groundcover with GLY, and then seeded in September 2010 with a 56-cm over-seeder (Bluebird International, Charlotte, NC). KB was seeded at a rate of 15 g m$^2$; CRF seeding rate was 34 g m$^2$. Intra-row groundcovers were hand broadcast and raked in. All groundcover plots were watered three times per week for six weeks. Inter-rows were periodically mown by the grower; intra-row groundcovers (CRF and NSC) were mown once during each growing season using a push mower or string trimmer. GLY was applied 2-3 times each season using a backpack sprayer at 1.12 kg ha$^{-1}$ acid equivalent, per grower standard practice. CG was initially applied at a depth of 7 cm in

June 2011. The material was obtained from MileStone Local Surfaces in Lincoln, NE and consisted of municipal waste glass which was mostly clear. The crushed glass shifted and settled more than expected, so the plots were renovated in March 2013. Clear glass was no longer available locally, and a replacement of mostly brown crushed glass was used. DG was applied at a rate of 1.6 kg m$^2$ in June 2011, reapplied in September 2011, and twice during each growing season in 2012 and 2013. This material was obtained from the University of Nebraska Department of Animal Science feed mill near Mead, Nebraska; typical chemical analysis of DG is 4.5% nitrogen, 1.1% phosphorus, and 1.3% potassium (*Blume, 2007*). Most data were collected in 2012 and 2013. Grapevine phenological stages were recorded using the Eichhorn-Lorenz scale, as described by *Coombe (1995)*.

## Soil temperature and chemical composition

Intra-row soil temperatures were measured using a digital thermometer with a 25.4 cm thermocouple probe (model HH508; Omega, Stamford, CT, USA). On each sampling date, the probe was inserted to its full depth twice for each plot (in the center of the vine row approximately 30 cm to the north and south of the center vine). Measurements were made between 1100 and 1300 CDT on May 28, 2012, April 5, 2013, and May 24, 2013. Soil sampling was conducted in NovemPber 2013. Two samples were collected from each intra-row and inter-row treatment, ignoring possible interactions between treatments. Each sample consisted of 3 soil cores to 20 cm depth, composited and submitted to Midwest Laboratories (Omaha, NE, USA) for analysis. Mean pH was 6.8 and organic matter (loss on ignition) was 3.9% (Data S1). DG appeared to dramatically increase nitrate, phosphorus, and potassium in the top 20 cm of soil, though statistical analysis was not conducted.

## Surface reflectance and canopy transmittance

Photosynthetically active radiation (PAR; 400-700 nm) was measured using a line quantum sensor (model LI-191SA; LI-COR Biosciences, Lincoln, NE, USA), and recorded by a data logger (model CR 23X; Campbell, Logan, UT, USA) between 1200 and 1400 CDT on July 25, 2012. This was a sunny day between veraison and harvest. Surface reflectance (reflected PAR) was measured with the quantum sensor inverted below the cordon. Transmitted PAR was measured with the quantum sensor face up, just below the cordon. Percent transmittances were calculated by dividing the transmitted PAR by the ambient PAR (measured above the canopy).

## Soil water content

Soil volumetric water content (VWC) was measured in the field, using a soil moisture meter (FieldScout TDR 300; Spectrum Technologies, Inc., Aurora, IL, USA) which utilizes time domain reflectometry. Measurements were taken using the standard soil setting; the instrument did not allow site-specific calibration. Readings of the average VWC in the top 0.2 m of soil, in both inter-rows and intra-rows of each plot, were repeated approximately every two weeks from bloom until veraison in 2012 and 2013. Each data point is an average of three measurements. Mulches were pushed aside before inserting the instrument, and replaced afterward.

### Stem water potential

Mid-day stem water potentials ($\Psi$) were measured using a pressure chamber (model 2005HGPL; Soil Moisture Equipment Corp., Santa Barbara, CA, USA), according to the method described by *Williams (2001)*. Measurements were repeated approximately every two weeks from bloom until veraison in 2012 and 2013 between 1200 and 1300 CDT. At each sampling date, one undamaged, fully expanded leaf on the center vine in each plot was wrapped in aluminum foil and sealed in a plastic zip-top sandwich bag to stop evapotranspiration for at least 60 min before sampling. Scissors were used to make a clean petiole cut, and $\Psi$ measurements were taken within 60 s of removing leaves from the vines.

### Weed cover

Intra-row percent weed cover was estimated using the Braun-Blanquet method to quantify visual observations of weed coverage (*Andújar et al., 2010*; *Wikum & Shanholtzer, 1978*). On three dates (September 7, 2011, June 25, 2012, and May 24, 2013), visual weed ratings for each plot were recorded by two observers for two randomly-placed 0.9 m² quadrats. The weed species most frequently observed were dandelion (*Taraxacum officinale* agg.), common lambsquarters (*Chenopodium album* L.), redroot pigweed (*Amaranthus retroflexus* L.), shepherd's purse (*Capsella bursa-pastoris* L.), goose grass (*Eleusine indica* [L.] Gaertn), and prostrate knotweed (*Polygonum aviculare* L.).

### Statistical analysis

Data were analyzed using SAS/STAT® Version 9.2 (SAS Institute, Cary, NC, USA). For each data set, a repeated measures analysis of variance (ANOVA) was implemented using the GLIMMIX procedure. Appropriate covariance structures were determined by fit statistics (smallest AICC). All two-way and three-way interactions were tested for significance in each ANOVA; significant main effects and interactions are reported in the results. All significance tests used $p < 0.05$.

## RESULTS & DISCUSSION

### Weather conditions during 2012 and 2013 growing seasons

Climate data were recorded at a High Plains Regional Climate Center weather station in David City, NE (13 km from the research site). Average daily minimum and maximum temperatures and total monthly precipitation for 2012 and 2013 growing seasons are presented in Table 1. In 2012 the spring and early summer were considerably warmer than in 2013, with total heat accumulation of 2101 and 1776 GDD (base 10 °C; average daily temperature method) during the growing seasons of 2012 and 2013, respectively. 2012 was also a drier growing season with marked differences in precipitation between March, May, July, and September of the two years.

### Soil temperature

Analysis of variance showed a significant interaction of treatment by date for intra-row soil temperatures (Table 2). In May of each year, the groundcover treatments lowered soil temperature in the top 25.4 cm, while mulch treatments raised soil temperature, relative

**Table 1  Mean daily maximum and minimum temperatures and total monthly precipitation during 2012 and 2013 growing seasons for David City, NE, 13 km from research site.** Each data point indicates the mean daily maximum and minimum temperatures and total monthly precipitation during vineyard floor management research.

| | Mean min air temp, 2012 (°C) | Mean min air temp, 2013 (°C) | Mean max air temp, 2012 (°C) | Mean max air temp, 2013 (°C) | Total precip, 2012 (mm) | Total precip, 2013 (mm) |
|---|---|---|---|---|---|---|
| March | 3.7 | −6.5 | 18.6 | 5.6 | 15 | 89 |
| April | 5.8 | −0.5 | 19.8 | 12.8 | 71 | 93 |
| May | 11.2 | 8.9 | 25.9 | 21.4 | 122 | 196 |
| June | 15.7 | 14.8 | 28.8 | 27.1 | 114 | 103 |
| July | 20.3 | 17.2 | 34.5 | 29.3 | 3 | 24 |
| Aug. | 15.2 | 17.8 | 30.6 | 28.9 | 44 | 44 |
| Sep. | 9.8 | 14.0 | 27.4 | 27.1 | 6 | 71 |
| Oct. | 2.0 | 3.9 | 16.2 | 17.0 | 37 | 125 |

**Table 2  Effects of intra-row vineyard floor management treatments on soil temperatures in a 'Marquette' vineyard planted in 2007 on a Hastings silty clay loam in Butler County, NE.** Each data point indicates the soil temperature measured under intra-row vineyard floor management treatments.

| Intra-row treatment | Date | Growth stage | Intra-row soil temperature (°C) | Standard error |
|---|---|---|---|---|
| CG | 28 May 2012 | 29 | 21.5 a | 0.13 |
| CRF | 28 May 2012 | 29 | 19.4 c | 0.13 |
| DG | 28 May 2012 | 29 | 21.7 a | 0.13 |
| GLY | 28 May 2012 | 29 | 20.1 b | 0.13 |
| NSC | 28 May 2012 | 29 | 19.5 c | 0.13 |
| CG | 05 April 2013 | 1 | 9.0 a | 0.10 |
| CRF | 05 April 2013 | 1 | 7.0 c | 0.10 |
| DG | 05 April 2013 | 1 | 7.6 b | 0.10 |
| GLY | 05 April 2013 | 1 | 7.5 b | 0.10 |
| NSC | 05 April 2013 | 1 | 7.4 b | 0.10 |
| CG | 24 May 2013 | 15 | 17.8 a | 0.10 |
| CRF | 24 May 2013 | 15 | 15.8 d | 0.10 |
| DG | 24 May 2013 | 15 | 16.5 b | 0.10 |
| GLY | 24 May 2013 | 15 | 16.2 c | 0.10 |
| NSC | 24 May 2013 | 15 | 15.7 d | 0.10 |

**Notes.**
Growth stages were estimated using the Eichhorn-Lorenz phenological scale, as described by . Measurements were taken to Coombe (1995) a depth of 25.4 cm. Within each date, means followed by the same letter are not significantly different at $p < 0.05$.
CG, crushed glass mulch; CRF, creeping red fescue; DG, distillers' grain mulch; GLY, glyphosate; NSC, non-sprayed control.

to bare soil (GLY). On April 5, 2013, CG had higher soil temperatures and CRF had lower soil temperatures than DG, NSC, and GLY plots, which did not differ.

In May of 2012 and 2013, CRF and NSC intra-row plots had lower soil temperatures than GLY plots. In April 2013, only CRF had lower soil temperature than GLY. The groundcover treatments may have reduced soil temperatures because of the evaporative demand of the vegetation. These data support previous research reporting lower soil temperature in

groundcover plots than in bare soil (*Van Huyssteen, Zyl & Koen, 1984*), although not all results have been consistent. In a California study, neither triticale nor rye groundcovers influenced soil temperature compared to tilled bare soil (*Steenwerth & Belina, 2008*).

In May 2012, CG and DG treatments had higher soil temperatures than GLY. In April 2013, only CG plots were warmer than GLY plots, while in May 2013, DG plots had higher soil temperature than GLY and CG soils were significantly warmer than those with DG. Mulches can have cooling or warming effects on the soil depending on the type of material used, because they change the radiation and energy balances at the soil surface. Mulches that reduce evaporation and shade the soil surface from incoming solar radiation generally have a cooling effect on soils (*Ham, Kluitenberg & Lamont, 1993*). White woven reflective material (*Sandler, Brock & Vanden Heuvel, 2009*) and crushed mussel shell mulch (*Creasy et al., 2007*; *Leal, 2007*) decreased soil temperatures relative to herbicide control, while mulching with gravel (*Nachtergaele, Poesen & Van Wesemael, 1998*) and silver reflective mulch (*Hutton & Handley, 2007*) increased soil temperatures. Researchers in Kansas concluded that mid-day soil temperatures were highest under mulches that efficiently transmit shortwave radiation while preventing the escape of longwave radiation (*Ham, Kluitenberg & Lamont, 1993*). This could explain the soil heating effect of the CG in this research. DG plots had higher soil temperatures in May of 2012 and 2013, converse to previous research which reported a buffering effect of organic mulches on soil temperature (*Mundy & Agnew, 2002*). It has been suggested that heat conduction between mulch and soil surface affects the ability of the mulch to moderate soil temperature (*Ham, Kluitenberg & Lamont, 1993*), which may be one reason why straw mulch had such a cooling effect (*Fourie & Freitag, 2010*) while DG increased soil temperatures in this study. A likelier explanation is that soil heating resulted from increased microbial activity under the DG mulch, as documented by *Boydston, Collins & Vaughn (2008)*.

Soil temperature can influence many aspects of vine growth and soil processes such as microbial respiration, organic matter decomposition rates and nutrient availability. Soil temperatures have been shown to affect cytokinin content in grapevine roots, percentage of buds that break, shoot length and dry weight, and yield (*Kliewer, 1975*). The percentage of buds that broke per vine increased with soil temperature, while leaf area, fruit set and cluster weights were highest at lower soil temperatures (*Kliewer, 1975*). Soil temperature may also affect bud-break timing, which could be important in cool climates where early-budding grape cultivars risk frost damage in spring (*Kliewer, 1975*; *Sandler, Brock & Vanden Heuvel, 2009*). However, the magnitude of differences in this research (approximately 2 °C difference between coolest and warmest treatments) probably was not sufficient to affect vine growth and no differences in bud-break or growth stage were observed between the treatments.

## Surface (vineyard floor) reflectance and canopy transmittance

The amount and type of reflected radiation depends on the properties of the soil surface, vegetation or mulch beneath vines (*Meinhold et al., 2010*). None of the treatments in this research affected the intra-row surface PAR reflectance on July 25, 2012 (Data S1; average reflectance over all treatments was 4.5% of incoming PAR). This result was unexpected

because there were obvious visual differences among the treatments. Many types of mulch have been found to increase PAR reflectance compared to bare soil, including crushed glass (*Mejias-Barrera, 2012*; *Ross, 2010*), and gravel (*Nachtergaele, Poesen & Van Wesemael, 1998*). Reflected radiation from the vineyard floor can be an important component of vines' total photosynthetically active radiation (PAR) interception, which can affect fruit composition (*Smart, Smith & Winchester, 1988*). Canopy density also influences vineyard floor surface reflectance, because a portion of the reflected light was originally transmitted through the canopy and thus alters the spectral quality of the incident light on the underlying surface. This is particularly applicable where vines are trained to a north-south trellis, since vines shade the intra-row area at mid-day (*Mejias-Barrera, 2012*). Because the vines' canopies were at full cover when PAR transmittance was measured (near veraison), transmitted light could have been saturated, obscuring possible differences due to canopy density and limiting the amount of light available to reflect. It is also possible that there could have been differences between the treatments which would have appeared if transmittance had been measured earlier in the season before canopy closure. A challenging aspect of reflective mulches is that they must remain clean and free of debris in order to function as intended (*Coventry et al., 2005*; *Sandler, Brock & Vanden Heuvel, 2009*). In this study, grass clippings blew onto the intra-row treatments following mowing of inter-rows, which could have masked differences in the surface reflectance. It is also possible that the quality of light, specifically the ratio of red to far-red radiation (which we did not measure), was altered by the reflective mulches; this ratio is involved in the regulation of phytochrome, which is important to many facets of grapevine growth and development (*Smart, Smith & Winchester, 1988*).

### Soil water content

The volumetric soil water content values measured in this study were unrealistically high due to calibration error; however, there were consistent treatment differences. We determined that investigation of relative differences would be appropriate. For each sampling date, the data were converted to ratios comparing each treatment mean to the grower standard treatment mean (RV for inter-rows and GLY for intra-rows) to describe the relationships between treatments.

In 2012, soil $VWC_{ratio}$ (0.2 m depth) in the inter-rows was affected by groundcover treatments on May 11 (KB < CRF < RV), June 12 (KB < RV) and July 10 [KB < (CRF = RV)]; on the other sampling dates, there were no differences in inter-row $VWC_{ratio}$ (Table 3). In 2013, KB had lower $VWC_{ratio}$ than RV on June 10; KB and CRF had lower $VWC_{ratio}$ than RV on July 3; KB had lower $VWC_{ratio}$ than both CRF and RV on July 12 and July 26; on June 28, the treatments did not differ. Plant species vary greatly in their evapotranspiration rates, and therefore their effects on soil moisture (*Lopes et al., 2004*). Differences in soil water content between the inter-row plots could have been influenced by stand density, rooting depth and water use of the groundcover species.

Intra-row soil moisture was highest for DG on each sampling date in 2013 and most dates in 2012 (Table 4). Surface-applied DG mulch absorbs water and reduces evaporation from the soil (*Blume, 2007*). A German study reported increased soil water content in

**Table 3** **Vineyard inter-row soil volumetric water content (VWC$_{ratio}$), expressed as a proportion of the VWC measured in grower standard treatment (resident vegetation; RV).** Vineyard floor management treatments applied to 'Marquette' grapevines planted in 2007 on a Hastings silty clay loam in Butler County, NE. Each data point indicates the soil volumetric water content of inter-row groundcover treatments, expressed as a proportion of the VWC measured under resident vegetation.

| Inter-row treatment | 2012 | | | 2013 | | |
|---|---|---|---|---|---|---|
| | Sampling date | Growth stage | VWC$_{ratio}$ | Sampling date | Growth stage | VWC$_{ratio}$ |
| KB | 5/11/2012 | 23 | 0.82 c | 6/10/2013 | 23 | 0.96 b |
| RV | 5/11/2012 | 23 | 1.00 a | 6/10/2013 | 23 | 1.00 a |
| CRF | 5/11/2012 | 23 | 0.92 b | 6/10/2013 | 23 | 0.99 ab |
| KB | 5/29/2012 | 29 | 0.97 a | 6/28/2013 | 29 | 0.96 a |
| RV | 5/29/2012 | 29 | 1.00 a | 6/28/2013 | 29 | 1.00 a |
| CRF | 5/29/2012 | 29 | 0.98 a | 6/28/2013 | 29 | 0.98 a |
| KB | 6/12/2012 | 32 | 0.88 b | 7/3/2013 | 31 | 0.88 b |
| RV | 6/12/2012 | 32 | 1.00 a | 7/3/2013 | 31 | 1.00 a |
| CRF | 6/12/2012 | 32 | 0.93 ab | 7/3/2013 | 31 | 0.93 b |
| KB | 6/25/2012 | 34 | 0.98 a | 7/12/2013 | 32 | 0.89 b |
| RV | 6/25/2012 | 34 | 1.00 a | 7/12/2013 | 32 | 1.00 a |
| CRF | 6/25/2012 | 34 | 0.98 a | 7/12/2013 | 32 | 0.99 a |
| KB | 7/10/2012 | 36 | 0.88 b | 7/26/2013 | 34 | 0.86 b |
| RV | 7/10/2012 | 36 | 1.00 a | 7/26/2013 | 34 | 1.00 a |
| CRF | 7/10/2012 | 36 | 0.99 a | 7/26/2013 | 34 | 0.98 a |
| KB | | | | 8/8/2013 | 36 | 0.93 b |
| RV | | | | 8/8/2013 | 36 | 1.00 a |
| CRF | | | | 8/8/2013 | 36 | 1.00 a |

**Notes.**
Growth stages were estimated using the Eichhorn-Lorenz phenological scale, as described by *Coombe (1995)*. Volumetric water content was measured using time domain reflectometry. Within each date, ratios followed by the same letter are not significantly different at $p = 0.05$.
KB, Kentucky bluegrass; CRF, creeping red fescue.

vineyard plots mulched with sawdust (*Huber et al., 2003*) which has a similar texture and particle size to the DG. *Mundy & Agnew (2002)* measured higher water content in vineyard soils under a variety of organic mulch treatments. Intra-row VWC was similar for CG and GLY on most dates in 2012, while in 2013 GLY had higher VWC than CG on all dates except June 10. Gravel mulch has been reported to increase evaporation from soil (*Nachtergaele, Poesen & Van Wesemael, 1998*). Conversely, research in New Zealand has found greater soil water content under CG compared to bare soil (*Ross, 2010*). On many dates in 2012 and 2013, CRF had the lowest VWC of all intra-row treatments. Although NSC generally had only slightly greater soil water content than CRF, NSC had higher VWC than CG on August 8 of 2013 (Table 4). Other studies have found lower soil water content under intra-row groundcover treatments, compared to herbicide controls (*Giese et al., 2015*; *Olmstead, 2006*), although in Iowa, intra-row CRF plots had higher VWC than herbicide plots (*Wasko, 2010*). Soil moisture is also important in the functioning of biological processes. Increased soil microbial activity was measured under 'Chardonnay' vines following application of
**Table 4** **Vineyard intra-row soil volumetric water content (VWC$_{ratio}$), expressed as a proportion of the VWC measured in grower standard treatment (glyphosate; GLY).** Vineyard floor management treatments applied to 'Marquette' grapevines planted in 2007 on a Hastings silty clay loam in Butler County, NE. Each data point indicates soil volumetric water content under intra-row ground cover treatments, expressed as a proportion of the VWC measured under glyphosate.

| Intra-row treatment | 2012 | | | 2013 | | |
| --- | --- | --- | --- | --- | --- | --- |
| | Sampling date | Growth stage | VWC$_{ratio}$ | Sampling date | Growth stage | VWC$_{ratio}$ |
| CG | 5/11/2012 | 23 | 0.97 a | 6/10/2013 | 23 | 0.94 b |
| CRF | 5/11/2012 | 23 | 0.67 c | 6/10/2013 | 23 | 0.93 b |
| DG | 5/11/2012 | 23 | 1.07 a | 6/10/2013 | 23 | 1.63 a |
| GLY | 5/11/2012 | 23 | 1.00 a | 6/10/2013 | 23 | 1.00 b |
| NSC | 5/11/2012 | 23 | 0.81 b | 6/10/2013 | 23 | 0.99 b |
| CG | 5/29/2012 | 29 | 1.02 b | 6/28/2013 | 29 | 0.86 c |
| CRF | 5/29/2012 | 29 | 0.86 d | 6/28/2013 | 29 | 0.64 d |
| DG | 5/29/2012 | 29 | 1.28 a | 6/28/2013 | 29 | 1.56 a |
| GLY | 5/29/2012 | 29 | 1.00 bc | 6/28/2013 | 29 | 1.00 b |
| NSC | 5/29/2012 | 29 | 0.93 cd | 6/28/2013 | 29 | 0.75 cd |
| CG | 6/12/2012 | 32 | 0.95 a | 7/3/2013 | 31 | 0.79 c |
| CRF | 6/12/2012 | 32 | 0.56 b | 7/3/2013 | 31 | 0.65 d |
| DG | 6/12/2012 | 32 | 1.09 a | 7/3/2013 | 31 | 1.55 a |
| GLY | 6/12/2012 | 32 | 1.00 a | 7/3/2013 | 31 | 1.00 b |
| NSC | 6/12/2012 | 32 | 0.60 b | 7/3/2013 | 31 | 0.71 cd |
| CG | 6/25/2012 | 34 | 0.91 b | 7/12/2013 | 32 | 0.74 c |
| CRF | 6/25/2012 | 34 | 0.79 c | 7/12/2013 | 32 | 0.59 d |
| DG | 6/25/2012 | 34 | 1.23 a | 7/12/2013 | 32 | 1.41 a |
| GLY | 6/25/2012 | 34 | 1.00 b | 7/12/2013 | 32 | 1.00 b |
| NSC | 6/25/2012 | 34 | 0.89 b | 7/12/2013 | 32 | 0.60 d |
| CG | 7/10/2012 | 36 | 1.08 ab | 7/26/2013 | 34 | 0.72 c |
| CRF | 7/10/2012 | 36 | 0.76 d | 7/26/2013 | 34 | 0.63 c |
| DG | 7/10/2012 | 36 | 1.17 a | 7/26/2013 | 34 | 1.14 a |
| GLY | 7/10/2012 | 36 | 1.00 bc | 7/26/2013 | 34 | 1.00 b |
| NSC | 7/10/2012 | 36 | 0.87 cd | 7/26/2013 | 34 | 0.68 c |
| CG | | | | 8/8/2013 | 36 | 0.68 d |
| CRF | | | | 8/8/2013 | 36 | 0.64 d |
| DG | | | | 8/8/2013 | 36 | 1.18 a |
| GLY | | | | 8/8/2013 | 36 | 1.00 b |
| NSC | | | | 8/8/2013 | 36 | 0.80 c |

**Notes.**

Growth stages were estimated using the Eichhorn-Lorenz phenological scale, as described by . Volumetric water content was measured using time domain reflectometry. Within each date, ratios followed by the same letter are not significantly different at $p = 0.05$.

CG, crushed glass mulch; CRF, creeping red fescue; DG, distillers' grain mulch; NSC, non-sprayed control.

mulched inter-row groundcover residue, which was attributed to higher soil water content in the mulched plots (*Jacometti, Wratten & Walter, 2007*).

Soil water content is an important variable affecting vine phenology and fruit yield and quality. Soil water availability affects rooting patterns and density of grapevines. Increasing

soil water content encourages root growth; root systems explored a much greater area in overhead sprinkler-irrigated vineyards than in drip-irrigated vineyards because there was a greater volume of moistened soil under sprinkler irrigation (*Soar & Loveys, 2007*). Vines grown under groundcover often have deeper roots because of reduced soil water content near the surface, where groundcover roots are more competitive (*Smart et al., 2006*). Too much moisture can contribute to excess vine vigor, while too little may restrict growth and yield (*Smart, 1985*). Excess vigor generates self-shading and a lack of vine balance (*Wheeler, Taylor & Young, 2008*). In regions where grapevines tend to produce overly vigorous vegetative growth, manipulation of soil water content may provide a useful method to correct the problem. Shoot growth, pruning weight and berry weight were increased by irrigation (*Hamman & Dami, 2000*). However, "excessive soil moisture" was observed under straw mulch in Iowa, which reduced berry size and likely delayed fruit ripening because of unlimited uptake of water after veraison (*Wasko, 2010*). Waterlogging can also restrict oxygen availability to roots, thereby reducing vine growth. Soil water status can affect fruit and wine quality parameters such as berry size, color and anthocyanins (*Williams & Matthews, 1990*). Mowing may be used to temporarily reduce groundcover water use, thereby making moisture more available to vines during key growth periods such as bloom and berry set (*Centinari et al., 2013*).

## Vine water potential

Mid-day stem water potentials ($\Psi$) were not affected by inter-row or intra-row treatments in either year of this study; overall means for each sampling date ranged from $-0.9$ to $-0.4$ MPa. These results support previous findings that intra-row groundcovers reduced soil moisture but did not significantly change predawn leaf $\Psi$, mid-day leaf $\Psi$ or xylem $\Psi$ (*Giese et al., 2015*). Olmstead and colleagues found reduced (more negative) vine water potential in some groundcover treatments, although the values were not indicative of water stress (*Olmstead et al., 2001*) which has commonly been defined as mid-day stem $\Psi$ values below $-1.3$ to $-1.6$ MPa (*Lovisolo et al., 2010*). Researchers in Oregon and California have reported little effect of inter-row groundcovers on leaf water potential (*Ingels et al., 2005*; *Schreiner & Sweet, 2005*; *Sweet & Schreiner, 2010*). In Virginia, intra-row creeping red fescue reduced stem $\Psi$ relative to herbicide strip, but vines in the groundcover plots did not show symptoms of water stress (*Hatch, Hickey & Wolf, 2011*). In our study VWC was measured only in the top 0.2 m of soil, while the roots of the grapevines likely penetrate much deeper(*Smart et al., 2006*). This is probably the main reason why the intra-row and inter-row treatments affected soil moisture but not stem $\Psi$.

In some cases, low (more negative) vine water potential has been associated with a reduction in vegetative vigor which is desirable in areas with abundant rainfall and soil fertility. *Hatch, Hickey & Wolf (2011)* found intra-row creeping red fescue decreased leaf layer number by 21% and pruning weight by 47%, in addition to reducing stem $\Psi$ in a Virginia 'Cabernet Sauvignon' vineyard (*Hatch, Hickey & Wolf, 2011*). A study in North Carolina reported slower shoot growth and reduced pruning weights as a result of intra-row groundcover although water potential differences were not significant (*Giese et*

**Table 5** **Effects of vineyard floor management treatments on intra-row percent weed cover in a 'Marquette' vineyard planted in 2007 on a Hastings silty clay loam in Butler County, NE.** Each data point indicates a Braun-Blanquet weed rating for intra-row vineyard treatments on a given date.

| Intra-row treatment | Date | Braun-Blanquet rating | Standard error |
|---|---|---|---|
| CG | 9/7/2011 | 0.0 c | 0.22 |
| CRF | 9/7/2011 | 1.2 b | 0.22 |
| DG | 9/7/2011 | 0.0 c | 0.22 |
| GLY | 9/7/2011 | 1.2 b | 0.23 |
| NSC | 9/7/2011 | 4.4 a | 0.22 |
| CG | 6/25/2012 | 1.4 de | 0.40 |
| CRF | 6/25/2012 | 1.0 e | 0.40 |
| DG | 6/25/2012 | 2.2 cd | 0.40 |
| GLY | 6/25/2012 | 2.6 abc | 0.42 |
| NSC | 6/25/2012 | 3.8 a | 0.40 |
| CG | 5/24/2013 | 0.7 b | 0.37 |
| CRF | 5/24/2013 | 1.7 b | 0.37 |
| DG | 5/24/2013 | 0.6 b | 0.37 |
| GLY | 5/24/2013 | 3.4 a | 0.39 |
| NSC | 5/24/2013 | 3.0 a | 0.37 |

Braun-Blanquet method for visual % weed cover estimation, as described by *Andújar et al. (2010)*:

| Braun-Blanquet rating | % weed cover | Braun-Blanquet rating | % weed cover |
|---|---|---|---|
| 0 | 0–1% | 3 | 26–50% |
| 1 | 1–10% | 4 | 52–75% |
| 2 | 11–25% | 5 | 76–100% |

Notes.

Within a date, means followed by the same letter are not significantly different at $p < 0.05$.

CG, crushed glass mulch; CRF, creeping red fescue; DG, distillers' grain; GLY, glyphosate; NSC, non-sprayed control.

*al., 2014*), which indicates vineyard floor vegetation affected grapevines in ways other than competition for water.

## Weed cover

Analysis of variance identified a significant interaction between intra-row treatment and date regarding weed cover. CG and DG provided complete weed control (0% weed cover) in September 2011 but did not control weeds as well in June 2012 or May 2013 (Table 5). This was likely a result of the time of year, as well as length of time since the treatments had been established. GLY weed control was progressively worse on each of the three dates, which was probably because of seasonal differences in weed seed germination and timing of herbicide application. The weeds in GLY plots were new seedlings that had germinated, rather than perennial weeds that were not controlled by the treatment.

Collectively, our results suggest that it is not advantageous to maintain a weed-free strip under the vine row in established vineyards where resources are not limiting. Other researchers have also determined that eliminating intra-row vegetation did not benefit established vineyards on deep, fertile soils (*Nonnecke et al., 2011*; *Wasko, 2010*) and that intra-row living mulches or groundcovers can reduce excessive vegetative vigor (*Giese*

*et al., 2014*; *Giese et al., 2015*; *Hatch, Hickey & Wolf, 2011*). By establishing a continuous vineyard floor groundcover or allowing resident vegetation to grow, growers can reduce their use of synthetic chemicals and provide habitat for arthropods and other wildlife.

## CONCLUSIONS

This study supports a growing body of evidence that maintenance of a weed-free strip under mature vines is unnecessary in vineyards situated on deep, fertile soils. Intra-row groundcovers can reduce growers' reliance on synthetic chemicals and reduce excess vegetative vigor. Mulches evaluated in this research successfully controlled weeds, but elimination of competition from weeds may not be desirable in fertile, vigorous vineyards.

For a potential vineyard site, if resident vegetation is appropriate (i.e., dominated by cool-season grasses) then it can be left in place as an inter-row groundcover. The vine rows can be disked or sprayed with herbicide in preparation for setting out new rooted cuttings. Even in areas where vigor reduction is desired for older vines, new plantings should be maintained with a weed-free strip or area around each vine in order to assist with establishment (*Bordelon & Weller, 1997*). Mulches present a sustainable alternative to GLY and other herbicides for this purpose. DG may be useful for first-year vineyards because it increases soil moisture retention and would only need to be applied one to two times to control weeds for the first season; thereafter, RV could be allowed to colonize the vine row, or an intra-row groundcover could be seeded. CG gave excellent weed control the first season, but because of transportation costs, labor, and logistics of application, it would only be practical if there is a local inexpensive source.

## ACKNOWLEDGEMENTS

The authors wish to acknowledge the technical assistance of Stephen Gamet, Charles Francis, Timothy Arkebauer, Elizabeth Walter-Shea, Benjamin Loseke, Vivian Shi, David Scoby, Mark Mesarch, and Erin Blankenship. We also thank the Bailey family at Fox Run Farms in Brainard, NE for hosting this research.

### Funding

Partial funding for this work was provided by the University of Nebraska Agricultural Research Division and the Nebraska Grape and Wine Board. There was no additional external funding received for this study. The funders had no role in study design, data collection and analysis, decision to publish, or preparation of the manuscript.

### Grant Disclosures

The following grant information was disclosed by the authors:
University of Nebraska Agricultural Research Division.
Nebraska Grape and Wine Boar.

## Competing Interests

The authors declare there are no competing interests.

## Author Contributions

- Christina M. Bavougian conceived and designed the experiments, performed the experiments, analyzed the data, prepared figures and/or tables, authored or reviewed drafts of the paper, approved the final draft.
- Paul E. Read conceived and designed the experiments, contributed reagents/materials/-analysis tools, authored or reviewed drafts of the paper, approved the final draft.

## Patent Disclosures

The following patent dependencies were disclosed by the authors:

David Blume 2007 #US7183237 B2

Method for the use of distiller's grain as herbicide and fertilizer

## Data Availability

The raw data are provided in a Data S1.

## Supplemental Information

Supplemental information for this article can be found online at http://dx.doi.org/10.7717/peerj.5082#supplemental-information.

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
