# Peer review of "Mulch and groundcover effects on soil temperature and moisture, surface reflectance, grapevine water potential, and vineyard weed management"

_PeerJ, doi:10.7717/peerj.5082_

## Round 0.1 · original submission · Major Revisions

· Academic Editor

Major Revisions

This is an interesting paper and both reviewers find it is worth publishing. However there are some concerns on the limitations of the experiment which can be explicitly described.

A diagram of the experimental design will be instructive.

I also have a question about the possible variation in soil type among treatments? There seems to be no soil analysis taken, at least for the site.

Measurement: Need to specific the depth of temperature measurement, surface?

Why do mulch treatments raise soil temperature? Organic mulch has a much lower thermal conductivity and thus should reduce temperature variation, i.e. lower soil temperature. Could it be the effect of time of measurement ?

Reviewer 1 ·

Basic reporting

Objectives well stated. Manuscript is well organized and very well written.

Experimental design

The design appears appropriate, although a strip-plot field design might have been more efficient. I do have some concerns about potential interaction among the main effects. These concerns are expressed in the "General Comments" section.

Validity of the findings

No comment.

Additional comments

This is a well organized and well written manuscript and should provide Midwestern winegrape growers with additional options for vineyard floor management. I have a few minor editorial suggestions and a few more substantive comments, as follows:

I’m unclear on the actual statistical field plot design given the nature of how data are presented in the tables. Authors state that a “randomized factorial design” with 5 intra- and 3-inter-row treatments were used. If a true factorial experiment was established, this would entail 5 x 3 (15) treatments, replicated 4 times = 60 field plots. I would then anticipate that statistical interpretations would include main effects (intra- and inter-row floor treatments), as well as their potential interactions. I failed to find mention of the potential inter- X intra-row interaction in the results or in the tabular data presented. The main effects are presented separately, so I must assume that there were no significant treatment interactions. This should be reported for each dataset.

Some of the methodology is unclear. For example, in the discussion of soil water content, we’re informed that each recorded value was an average of three measurements (Line 163). Is that three measures per treatment replicate (3 X 4 reps)? Similarly, how many leaves were sampled for mid-day stem water potential?

Authors mention at lines 246-247 that the lack of treatment effects on reflected PAR was unexpected “because there were obvious visual differences among the treatments.” This is also mentioned in the abstract, yet we’re not told how those differences were visually identified. What is the value of the subjective comments?

The ecophysiological measures that were taken were helpful in explaining potential vine responses, but the obvious potential responses of vine size (cane pruning weight), shoot vigor, components of yield, and fruit chemistry are all lacking. Presumably those responses are discussed in a companion paper, but it would be more satisfying to see the vine responses in this same paper with the data presented here.

Some very minor issues: the work of Giese et al. of 2014 (lines 338-340) was conducted in the Yadkin Valley of NC, not in Virginia. The soil temperature data in Table 2 should have a footnote defining the depth at which the soil temperature was measured. Also – the similarity of the standard errors reported here is suspicious. Line 40: “none of the treatments affected mid-day vine water potential”. How do you know that all treatments didn’t affect mid-day vine water potential the same way? In reality, Table 5 is probably not needed – the results could be discussed in the narrative.

Reviewer 2 ·

Basic reporting

Mulch and groundcover effects on soil temperature and moisture, surface reflectance, grapevine water potential, and vineyard weed management (#22845)

Line 50-52: How does maintaining permanent sod cover between vineyard rows… contribute to the sustainability of vineyard agroecosystems?
Line 60-62: REF?
Line 72-73: RE-word/DELETE: Excess vine vigor also creates a lack of vine balance requiring
requires high labor inputs.
Line 75: REF? especially regarding time requirements to prune and position shoots
Line 77: the phrase: “vine balance” is open to wide interpretation. In this sentence, the phrase “restore vine balance” might be replaced with the phrase: “reduce excessive vine vigor”
Line 88: REF?
Line 95: REF?
Line 108: own-rooted Marquette vines?
Line 115-121: it is not clear to me if the 5 under-trellis treatments had a single and consistent
inter-row treatment or if each under-trellis treatment was combined with each of the 3 inter-row treatments. Or…were each of the 3 inter-row treatments maintained with a consistent under-trellis treatment?
Line 122-137: What is the width of glyphosate treated strip and the width of the intra-row ground cover treatments?
Line 190-196: Could calculated GDDs be added in order to allow comparison with other regions?
Line 211: DELETE: “both”
Lines 231-241: At what soil temperature can one expect to see impacts on vine growth? How do these temperatures compare with your reported soil temperatures?
Line 239: DELETE “this” and replace with “our”
Line 245: A single date measurement of PAR likely did not provide an adequate depiction of field conditions or treatment effect
Lines 253: and thus alters the spectral quality of the incident light on the underlying surface is altered
Line 255: DELETE “since” which indicates a time past…and replace with “because”
Line 269: why wasn’t a gravimetric measurement of soil water provided? It could have been used
as a benchmark and as calibration reference for possible interpretation of the soil water
measurements collected with the TDR probe after the fact
Line 304: this is a confusing sentence…is drip irrigation necessarily associated with less soil water content
compared to sprinkler irrigation? If this was the case in the cited research, then state that fact.
Line 320-321: DELETE: Table 5, no reason to include a table that does not contain differences or unusually high or low vine water status.
Line 332-333: appropriate discussion statement…any reference to support the likelihood of deeper grapevine roots?
Line 338: DELETE: “Virginia” and replace with “North Carolina”, the referenced study was conducted in North Carolina
Line 341: What “other” ways may groundcovers compete with grapevines?
Line 355: Again…I suggest to defer from the use of “improve vine balance” as an effect or outcome…”reducing vine vigor” is more exact or accurate

CONCLUSION:
** Although data was collected on groundcover treatment impact on light interception by the vine canopy (PAR), soil moisture, vine water potential and soil temperature, these parameters were not correlated to vine growth or yield. Groundcover biomass or weed % may have correlated to vine canopy density or pruning weights. Why was vine nitrogen status not measured?**

Line 362-364: This is a broad statement that is not supported by your research…you show no pruning weight or yield data that would support it. How can you be sure that the glyphosate intra-row strip is “unnecessary”, if this yield and pruning weight data…is not shown or presented? Most growers would be want to know the effect of a vineyard practice on yield.
Line 363-365: the reduction of vineyard labor was not directly supported/substantiated by your
results…did you measure it? Was “vine balance” measured? Overall, I recommend refraining from the use of the terms “vine balance”
Line 367-369: DELETE: Cool- season grasses are the most practical inter-row groundcovers because they resume growth early in the spring, which facilitates vineyard management tasks by providing a firm footing for tractor and foot traffic. This was not a measured finding of your research.

Experimental design

no comment

Validity of the findings

Mulch and groundcover effects on soil temperature and moisture, surface reflectance, grapevine water potential, and vineyard weed management (#22845)

Line 50-52: How does maintaining permanent sod cover between vineyard rows… contribute to the sustainability of vineyard agroecosystems?
Line 60-62: REF?
Line 72-73: RE-word/DELETE: Excess vine vigor also creates a lack of vine balance requiring
requires high labor inputs.
Line 75: REF? especially regarding time requirements to prune and position shoots
Line 77: the phrase: “vine balance” is open to wide interpretation. In this sentence, the phrase “restore vine balance” might be replaced with the phrase: “reduce excessive vine vigor”
Line 88: REF?
Line 95: REF?
Line 108: own-rooted Marquette vines?
Line 115-121: it is not clear to me if the 5 under-trellis treatments had a single and consistent
inter-row treatment or if each under-trellis treatment was combined with each of the 3 inter-row treatments. Or…were each of the 3 inter-row treatments maintained with a consistent under-trellis treatment?
Line 122-137: What is the width of glyphosate treated strip and the width of the intra-row ground cover treatments?
Line 190-196: Could calculated GDDs be added in order to allow comparison with other regions?
Line 211: DELETE: “both”
Lines 231-241: At what soil temperature can one expect to see impacts on vine growth? How do these temperatures compare with your reported soil temperatures?
Line 239: DELETE “this” and replace with “our”
Line 245: A single date measurement of PAR likely did not provide an adequate depiction of field conditions or treatment effect
Lines 253: and thus alters the spectral quality of the incident light on the underlying surface is altered
Line 255: DELETE “since” which indicates a time past…and replace with “because”
Line 269: why wasn’t a gravimetric measurement of soil water provided? It could have been used
as a benchmark and as calibration reference for possible interpretation of the soil water
measurements collected with the TDR probe after the fact
Line 304: this is a confusing sentence…is drip irrigation necessarily associated with less soil water content
compared to sprinkler irrigation? If this was the case in the cited research, then state that fact.
Line 320-321: DELETE: Table 5, no reason to include a table that does not contain differences or unusually high or low vine water status.
Line 332-333: appropriate discussion statement…any reference to support the likelihood of deeper grapevine roots?
Line 338: DELETE: “Virginia” and replace with “North Carolina”, the referenced study was conducted in North Carolina
Line 341: What “other” ways may groundcovers compete with grapevines?
Line 355: Again…I suggest to defer from the use of “improve vine balance” as an effect or outcome…”reducing vine vigor” is more exact or accurate

CONCLUSION:
** Although data was collected on groundcover treatment impact on light interception by the vine canopy (PAR), soil moisture, vine water potential and soil temperature, these parameters were not correlated to vine growth or yield. Groundcover biomass or weed % may have correlated to vine canopy density or pruning weights. Why was vine nitrogen status not measured?**

Line 362-364: This is a broad statement that is not supported by your research…you show no pruning weight or yield data that would support it. How can you be sure that the glyphosate intra-row strip is “unnecessary”, if this yield and pruning weight data…is not shown or presented? Most growers would be want to know the effect of a vineyard practice on yield.
Line 363-365: the reduction of vineyard labor was not directly supported/substantiated by your
results…did you measure it? Was “vine balance” measured? Overall, I recommend refraining from the use of the terms “vine balance”
Line 367-369: DELETE: Cool- season grasses are the most practical inter-row groundcovers because they resume growth early in the spring, which facilitates vineyard management tasks by providing a firm footing for tractor and foot traffic. This was not a measured finding of your research.

---

## Round 0.2 · accepted · Accept

· Academic Editor

Accept

The manuscript has been revised according to the reviewers' comments. There are still few typos that need to be checked in the production process

Reviewer 1 ·

Basic reporting

no further comments

Experimental design

no further comments

Validity of the findings

no further comments

Additional comments

no further comments